# CROSS DOMAIN VULNERABILITY DETECTION USING GRAPH CONTRASTIVE LEARNING

**Mahmoud Zamani, Saquib Irtiza, Shamila C. Wickramasuriya, Latifur Khan & Kevin W. Hamlen**
Department of Computer Science
The University of Texas at Dallas
Richardson, TX 75080, USA
`{mahmoud.zamani, saquib.irtiza, scw130030, lkhan, hamlen}@utdallas.edu`

## ABSTRACT

A new approach to software vulnerability detection is proposed and evaluated, which combines state-of-the-art contrastive learning (CL) via GraphCL with a new cross-domain control property graph (CDCPG) model that combines source- and binary-level code features. Self-supervised learning (SSL), including CL, is critical for addressing the longstanding difficulty of building large, high-quality data sets for this domain. The proposed method trains on a new graph dataset generated from code repositories of six widely used C/C++ applications. The combination of source and binary features affords detection of vulnerabilities that are invisible at only one level of granularity. Experiments using different augmentation techniques and loss functions to show that GraphCL with CDCPG performs better than any other evaluated detection strategy in many scenarios.

## 1 INTRODUCTION

Automatic detection of software vulnerabilities is a high interest cybersecurity research topic because of the increasing infeasibility of fully auditing large software systems manually. Undetected vulnerabilities in critical systems can cause significant damage to organization reputation and integrity, even including loss of life. As software complexity and evolution rates have increased, the difficulty of finding all exploitable vulnerabilities before the software is deployed has escalated, opening more doors for threat actors to launch successful attacks against live assets.

Poor true positive and true negative rates of traditional methods (Li et al., 2019; Xu et al., 2017) and the cost of manually extracting features for machine learning models (Nguyen & Tran, 2010; Neuhaus et al., 2007) have motivated automatic feature extraction using deep learning techniques (Li et al., 2018; 2021). Despite their improvement upon prior approaches, these methods overlook the control-flow, dataflow and syntactic structure of programs. Instead, they consider programs as natural language instances, which lose many features critical for identifying certain vulnerability classes. To overcome this issue, subsequent works (Zhou et al., 2019; Chakraborty et al., 2022) propose graph learning on *code property graphs* (CPGs) (Yamaguchi et al., 2014), which integrate *abstract syntax trees* (ASTs), *control-flow graphs* (CFGs), and *data dependency graphs* (DDGs) to incorporate syntactic and semantic information. Unfortunately, CPGs only include source-level information, preventing detection of vulnerabilities that can only be seen at the binary level, such as those introduced during compilation.

To overcome this limitation, we explore a new kind of CPG that includes binary file (BIN) information and its relation to source-level CPG features. The binary features augment the source features by revealing compiler decisions and results of compiler analyses (e.g., abstraction of function and variable names, reordering of stack variables, etc.) that cannot be inferred from source code alone and can lead to new vulnerabilities only detectable at binary-level. Our new *cross-domain code property graphs* (CDCPGs) can identify vulnerabilities both in the source code and its corresponding compiled binary code. Figure 2 in §A.1 shows a sample CDCPG. Also, most existing methods use supervised learning models which require a large volume of labeled data to train. Since it is very expensive to collect such a large dataset in this domain, we use a state-of-the-art self-supervised learning (SSL), GraphCL, to evaluate our curated dataset. Our key contributions are:

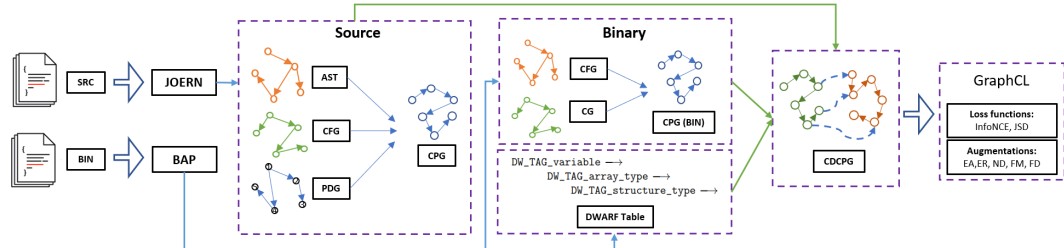

Figure 1: Data collection along with graph contrastive learning technique. More details in §A.3.

- We develop a new graph type called CDCPG that combines both binary-level code and source-code level features. We collect graphs for six widely used open-source programs into a new curated dataset for vulnerability detection. Additionally, the dataset contains other graph types such as AST, CPG and CFG.
- We evaluated the performance of CDCPG on a contrastive learning model, GraphCL, to see whether it performs better than other graph types. We also use different loss functions and augmentation techniques to identify the best performing combination for this algorithm.

## 2 DATA COLLECTION AND PROPOSED METHOD

We consider open-source software repositories of six different applications, from which we collect functions marked as vulnerable according to the Common Vulnerabilities and Exposures (CVE) database (MITRE, 2023) and their corresponding binary code from compiled binaries. Then we use Joern (Yamaguchi et al., 2014), a C/C++ analysis framework, to generate AST, CFG, and CPG graphs from the source codes. We also generate BIN graphs from the corresponding binary code using the Binary Analysis Platform (BAP) (Brumley et al., 2011). The resulting graphs are combined into a CDCPG by introducing new edge types that relate syntactic and semantic features of both domains that correspond. Next, we train GraphCL on these graphs using various loss functions (InfoNCE and Jensen-Shannon Divergence (JSD)) and augmentation techniques (edge addition (EA), edge removal (ER), node dropping (ND), feature masking (FM), and feature dropping (FD)) to determine the best performing combination. Figure 1 summarizes and §A.3 details the pipeline.

## 3 EVALUATION

Our evaluation uses F1-micro and F1-macro metrics to account for data imbalance. F1-micro gives equal weight to all the instances, whereas F1-macro gives equal weight to each class. This allows both minority and majority classes to contribute equally towards F1-macro. Our experiments are conducted on each application separately for all the graph types, loss functions, and augmentation techniques mentioned in §2. Table 1 shows the results.

Table 1: F1-scores using ND augmentation and JSD loss function with GraphCL algorithm on all graphs. Results show that CDCPG outperforms other graph types for the TCPDump application.

| Graph Type | AST | CFG | CPG | BIN | CDCPG |
|---|---|---|---|---|---|
| **F1-Micro** | 0.85 | 0.63 | 0.63 | 0.63 | **1.00** |
| **F1-Macro** | 0.46 | 0.39 | 0.39 | 0.39 | **1.00** |

## 4 CONCLUSION

Combining source-level and binary-level code data into a single graph called CDCPG facilitates detection of vulnerabilities at both levels. Contrastive learning with multiple graph augmentation methods and loss functions indicates that the approach is more effective than prior approaches.

## ACKNOWLEDGMENTS

This research was supported in part by DARPA Award N6600121C4024 and ARO Award W911NF2110032.

## URM STATEMENT

The authors acknowledge that at least one key author of this work meets the URM criteria of ICLR 2023 Tiny Papers Track.

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

## A    APPENDIX

### A.1    SAMPLE OF CDCPG GRAPH

Figure 2 shows a sample CDCPG graph generated for the code snippet in Figure 3. The paths labeled with blue and grey arrows belong to the CPG generated from the source-level code, whereas the path represented by black arrows are the ones that belong to the CPG of the binary-level code. The dotted purple lines show the mapping between the different components in the source and binary level subgraphs.

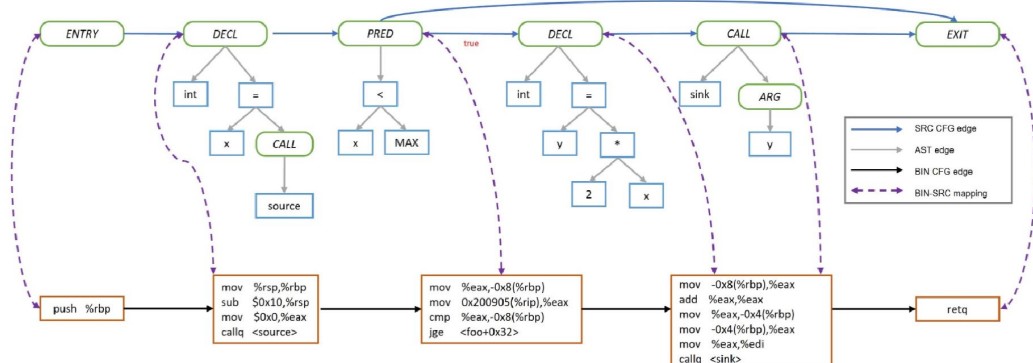

Figure 2: Cross-Domain Code Property Graph (CDCPG) generated from the snippet in Figure 3.

```
void foo(){
    int x = source();
    if (x < MAX){
        int y = 2  x;
        sink(y);
    }
}
```

Figure 3: Code Snippet from which the CDCPG in Figure 2 is generated.

## A.2 ADDITIONAL EXPERIMENTS

We conducted further evaluation on just CDCPG graphs using different augmentation techniques and loss functions to find the optimal parameters for GraphCL on the vulnerability detection task. Table 2 shows the scores averaged across all the six applications in the dataset. The highest scores for each augmentation mode are in bold. JSD is the better performing loss function because it gives the highest score for three out of the five augmentation techniques using both the metrics. Node Dropping (ND) is the best performing augmentation technique for JSD loss, since it gives the highest scores for both F1-Micro and F1-Macro.

Table 2: Evaluation of different augmentations and loss functions using CDCPG for all applications

| Mode | InfoNCE-F1Mi | InfoNCE-F1Ma | JSD-F1Mi | JSD-F1Ma |
|------|--------------|--------------|----------|----------|
| EA | 0.852 | 0.522 | **0.863** | **0.565** |
| ER | **0.843** | 0.585 | 0.825 | **0.600** |
| ND | 0.738 | 0.522 | **0.885** | **0.650** |
| FM | 0.843 | **0.522** | **0.883** | 0.468 |
| FD | **0.875** | **0.555** | 0.768 | 0.425 |

## A.3 DATASET DESCRIPTION

Tables 3 and 4 summarize our dataset, which spans six applications popular in the open source community, including many with security-critical functionalities. We selected these applications based on the number of security issues reported against them and also depending on the importance of the applications. We generated five types of graphs (AST, CFG, CPG, BIN, and CDCPG) for all the applications.

BAP generates CFGs from binary code, which we combine with Call Graphs (CGs) to facilitate analysis between different functions. Binary attributes are extracted from the combined graph to form the binary CPGs. Finally, to determine the relationship between executable and original source code, a Debug With Arbitrary Record Format (DWARF) table is generated during compilation.

Table 3: List of target applications and their descriptions

| Application | Description |
|---|---|
| Sudo | Delegates security privileges to other users or tasks |
| Poftpd | Ftp server with configurable features |
| Libtiff | Tagged Image File Format (TIFF) library |
| Libpng | Portable Network Graphics (PNG) library |
| Freetype | Renders text into bitmaps |
| TinTin | Console telnet client for online gaming |
| Tcpdump | Data network packet analyzer |
| OpenSSH | Secure networking utility for Secure Shell protocol |

Table 4: App dataset. The last two columns report vulnerable and non-vulnerable functions, resp.

| App Name | Graph Types | Edges | Nodes | Graphs | Vul. | Non-Vul. |
|---|---|---|---|---|---|---|
| **LibPNG** | CPG | 1845382 | 82413 | 324 | 25 | 299 |
| | CFG | 3104607 | 12487 | | | |
| | BIN | 7861098 | 28248 | | | |
| | AST | 2628710 | 81777 | | | |
| | CDCPG | 8415974 | 116839 | | | |
| **LibTIFF** | CPG | 5125862 | 79009 | 438 | 33 | 405 |
| | CFG | 4498837 | 13142 | | | |
| | BIN | 9156127 | 47104 | | | |
| | AST | 3679494 | 78132 | | | |
| | CDCPG | 5746318 | 116839 | | | |
| **TCPDump** | CPG | 2074116 | 33461 | 100 | 15 | 85 |
| | CFG | 328240 | 6018 | 100 | 15 | 85 |
| | BIN | 607635 | 13679 | 100 | 15 | 85 |
| | AST | 6092203 | 57465 | 200 | 26 | 174 |
| | CDCPG | 4017677 | 48944 | 100 | 15 | 85 |
| **Sudo** | CPG | 5636108 | 41858 | 187 | 9 | 178 |
| | CFG | 740870 | 6213 | | | |
| | BIN | 2664002 | 25731 | | | |
| | AST | 4384746 | 41506 | | | |
| | CDCPG | 1617438 | 71463 | | | |
| **TinTin** | CPG | 8764300 | 44903 | 270 | 4 | 266 |
| | CFG | 1006857 | 6636 | | | |
| | BIN | 3747693 | 24376 | | | |
| | AST | 586926 | 44363 | | | |
| | CDCPG | 2361790 | 78414 | | | |
| **OpenSSH** | CPG | 5750707 | 70560 | 100 | 21 | 79 |
| | CFG | 732305 | 1003 | | | |
| | BIN | 1965966 | 36199 | | | |
| | AST | 4186296 | 70352 | | | |
| | CDCPG | 10937966 | 112250 | | | |

