# OpenReview forum: "Cross Domain Vulnerability Detection using Graph Contrastive Learning"
_ICLR.cc/2023/TinyPapers — Submitted to Tiny Papers @ ICLR 2023_

### Official Review · Reviewer_fGpG · 2023-03-29

**Confidence:** 3

**Summary Of Contributions:**

The authors evaluate the performance of GraphCL, a state-of-the-art Contrastive Learning (CL) method for Self-Supervised Learning (SSL) in vulnerability detection. They introduce a custom dataset featuring a new graph structure called Cross Domain Control Property Graph (CDCPG), which combines code-level and binary-level CPG graphs for improved detection.

**Rating:**

High Potential (HP): a submission which meets the reviewing criteria and has potential to make an impact on the field

**Strengths And Weaknesses:**

Strengths:

1. The creation of a novel CDCPG dataset offers a unique and more comprehensive representation for vulnerability detection tasks.

2. This thorough investigation helps identify the best combination to mitigate class imbalance and improve model performance.

3. The study contributes valuable insights into SSL methods for addressing the challenge of limited labeled data in the field of vulnerability detection.

**Suggested Changes:**

None

---

> ### Author Response · Authors · 2023-04-25
> **Gratitude to the reviewer**
>
> We greatly appreciate your feedback and would like to thank you for recommending our work as having potential to have an impact on the field.

---

### Official Review · Reviewer_epGw · 2023-03-30

**Confidence:** 3

**Summary Of Contributions:**

This work generate a custom dataset with graphs to evalute the model performance. The state-of-the-art method is the contrastive learning for vulnerability detection task. However, the relationship between vulnerability detection and GraphCL is not well explained.

**Rating:**

Great Start (GS): a submission which meets some of the reviewing criteria but has room for improvement

**Strengths And Weaknesses:**

Strength:
•	Authors proposed a new graph dataset to detect vulnerabilities using GraphCL, which adopts graph augmentation methods and evaluates the result based on different loss functions.

Weakness:
•	The relationship between vulnerability detection and GraphCL is not well explained.

•	The task is not clear, i.e., is framework proposed for training a better and robustness model?

•	The contribution is not well stated.


**Suggested Changes:**

•	What does the meaning of the code-level and binarylevel CPG graphs?

•	The motivation and the task need to be polished.

---

> ### Author Response · Authors · 2023-04-25
> **We have addressed the reviewers' concern about the contributions and the motivation of the paper and have also clarified what code-level and binary-level CPG means.**
>
> Thank you for taking your time to review our paper. We have revised our paper based on the reviews that you have provided and have tried to address all the weaknesses and questions.
>
> •	Q. The relationship between vulnerability detection and GraphCL is not well explained.
> A.	We have observed that most existing methods in vulnerability detection use supervised learning models which require a large volume of labeled data to train. Since it is very expensive to collect such a large dataset in this domain, we use state-of-the-art Self-Supervised Learning (SSL) method, particularly GraphCL, to evaluate our curated dataset.
>
> •	Q. The task is not clear, i.e., is framework proposed for training a better and robust model? The motivation and the task need to be polished.
>
> A.	Existing works in this domain that use graph learning techniques require source code to generate the graphs which is often not publicly available. Instead, only binary (.bin) and executable (.exe) files are shared. The main goal of the paper is to propose a new framework that we can use to generate a new graph type, such that we can detect vulnerabilities even if the source code is not available. Also, during compilation, additional complexities such as abstraction of function and variable names, reordering of stack variables etc. are often introduced that might lead to new vulnerabilities  only detectable at binary-level. That is where our new custom graph type called Cross Domain Control Property Graph (CDCPG), plays a crucial role. It can identify vulnerabilities both in the source code and its corresponding compiled binary code which was not possible before.
>
> •	Q. The contribution is not well stated.
>
> A.	The main contributions are as follows:
> 1.	We developed a new graph type called CDCPG that combines both binary-level code and source-code level features. We collect graphs for six widely used open-source programs into a new curated dataset for vulnerability detection. Additionally, the dataset contains other graph types such as AST, CPG and CFG.
> 2.	We evaluate the performance of CDCPG on a contrastive learning model, GraphCL, to see whether it performs better than other graph types. We also use different loss functions and augmentation techniques to identify the best performing combination for this algorithm.
>
> •	Q. What is the meaning of the code-level and binary-level CPG graphs?
>
> A.	Code-level CPG graphs are those CPG graphs that are generated from the source code of the applications using an application called JOERN. Binary-level CPG graphs on the other hand are the CPG graphs that are generated by BAP. When these two CPG graphs are combined, we get our new custom graph type which we call CDCPG. This is different from normal CPGs because it combines the features from both source code and binary code. This has never been done before in this domain.

---

### Official Review · Reviewer_WmBe · 2023-04-03

**Confidence:** 4

**Summary Of Contributions:**

Developed Cross Domain Control Property Graph (CDCPG). And used GraphGL to evaluate the proposed graph dataset.

**Rating:**

Needs Clarification (NC): a submission which does not meet the reviewing criteria and needs clarification for its described problem or solution

**Strengths And Weaknesses:**

## Strengths:
### 1. The motivation of this paper is sound: "There are projects which their source codes are not available and the owners did not make the projects open-source".

## Weaknesses:
### 1. The writing of this paper is not good enough. There are **many grammar mistakes or typos** in the paper.
For example, in the Abstract: 'in the process find' should be 'in the process of finding', in the Introduction Section: 'import role' should be 'important role', and in the Proposed Method Section, 'performance compare with', should be 'performance compared with. 'And most of the nouns lack articles.

### 2. Did not write clearly how to generate the CDCPG graph. Figure 1 is not referred to in the text, and no explanation of the process of generation.
Since it is the most crucial part of this paper, it should be clearly explained in the text.


**Suggested Changes:**

1. Should explain how to generate CDCPG from six graph datasets.

2. Should explain how to evaluate the quality of the CDCPG.

---

> ### Author Response · Authors · 2023-04-25
> **We have fixed the syntactic and grammatical errors in the paper and have also conducted additional experiments to clarify the doubts that the reviewer had.**
>
> Thank you for taking your time to review our paper. We apologize for the grammatical mistakes that we have made in the paper. We addressed all such problems in our revised version and hope that it is comprehensible now.
>
> •	In response to your question on how we generate the CDCPG graph dataset, we initially go over open-source software repositories of six different applications and collect functions from them that are marked as vulnerable according to Common Vulnerabilities and Exposures (CVE). We select these applications based on the number of security issues reported against them and depending on the importance of the applications. We also collect the corresponding binary code for those functions.
>
> Then we use JOERN, a C/C++ analysis framework to generate AST, CFG and CPG graphs from the source level codes. We also generate binary CFG from the corresponding binary code using a tool called Binary Analysis Platform (BAP). BAP also generates Call Graphs (CG) that are combined with CFG to facilitate analysis between different functions. CGs are graphs that represent the calling relationship between different call events in a program. Next, binary attributes are extracted from this new combined graph to form the binary-level CPG graphs. Finally, to determine the relationship between executable and original source code, the Debug With Arbitrary Record Format (DWARF) table is generated during compilation that helps us to link different components of source-level graph with binary-level CPG graphs. Once the links are established, we obtain our CDCPG graphs.
>
> •	To evaluate the quality of CDCPG, we conducted experiments on the graphs generated from TCPDump application. The results are reported as follows:
>
> AST -> F1-Micro 0.85, F1-Macro 0.46
>
> CFG -> F1-Micro 0.63, F1-Macro 0.39
>
> CPG -> F1-Micro 0.63, F1-Macro 0.39
>
> BIN -> F1-Micro 0.63, F1-Macro 0.39
>
> CDCPG -> F1-Micro 1.00, F1-Macro 1.00
>
> It is evident that our custom graph type CDCPG outperforms other graph types in this scenario 	and can detect all the vulnerabilities. This shows that CDCPG is effective in detecting 	vulnerabilities in scenarios where other graph types fail.

---

### Meta-Review · Area_Chair_9GNS · 2023-04-04

**Recommendation:** Invite to revise
**Confidence:** 4

**Metareview:**

Good paper with a clear motivation. It could be better if suggestions from reviewers could be incorporated into the manuscript. Especially, the writing could be further improved.

**Summary:**

The authors assess the efficacy of GraphCL, a cutting-edge Contrastive Learning (CL) technique for Self-Supervised Learning (SSL) in detecting vulnerabilities. They present a tailored dataset with a novel graph structure termed Cross Domain Control Property Graph (CDCPG), which merges code-level and binary-level CPG graphs to enhance detection capabilities. Further clarifications are needed for certain parts of the method.

**Reason For Not Giving A Higher Recommendation:**

Further clarifications are needed for certain parts of the method.

**Reason For Not Giving A Lower Recommendation:**

N/A

---

### Decision · Program_Chairs · 2023-04-07

Revision accepted; invite to archive